# Available Flood Evacuation Time for High-Risk Areas in the Middle Reach of Chao Phraya River Basin

**Sarawut Jamrussri [1,*] and Yuji Toda [2]**

[1]    Electricity Generating Authority of Thailand (EGAT), Bang Kruai, Nonthaburi 11000, Thailand
[2]    Graduate School of Engineering, Nagoya University, Furo-cho, Chikusa-ku, Nagoya 464-8603, Japan;
        ytoda@cc.nagoya-u.ac.jp
*    Correspondence: sarawut.jam@egat.co.th; Tel.: +66-97-201-3165

**Abstract:** Information about risk is essential to design flood risk management programs. To our knowledge, this is the first attempt to develop an emergency flood evacuation plan based on flood risk assessment. Flood risk assessment in the middle Chao Phraya River Basin (CPRB) was simultaneously analyzed and mapped as the product of flood hazard, and social vulnerability maps generated by fuzzy Analytic Hierarchy Process (AHP) and fuzzy logic. One of the purposes of flood risk mapping is to promote proper and prompt evacuation actions for residents. The emergency flood evacuation model was tested to explore the available time of evacuation, to reduce the risk or even the loss of life. The simulation results showed that significant time was available for evacuation in the middle CPRB. This was calculated based on a physical status of evacuees, safe evacuation condition, shortest evacuation path, flood shelter, and road capacity.

**Keywords:** flood hazard map; flood evacuation strategies; middle Chao Phraya River Basin

## 1. Introduction

In Thailand, half of the natural disasters recorded during 1983 and 2012 were caused by floods, and their impacts were the largest in terms of the amount of economic damage [1]. The 2011 flood in the Chao Phraya River Basin (CPRB) was a very remarkable flood disaster in Thailand, resulting in damages and losses amounting to US $46.5 billion [2]. Millions of people were evacuated and more than 1000 residents lost their lives. The CPRB is a flood-prone area because of its topographical characteristics. The various extensive flood countermeasures have been taken to control the magnitude and frequency of flood, and also to protect citizens and properties in this Basin. However, the CPRB is still under threat of flood risk.

Comprehensive flood risk assessment has been widely applied to provide information regarding the risk of implementing various types of flood measures, such as floodplains management and flood evacuation [3,4]. Basically, the degree of flood risk levels usually dependst on two main aspects: The first aspect is related to flood events, flood hazard; and the second one is related to people, social vulnerability. Flood risk assessment is often established based on the concept of flood hazard and social vulnerability assessment [5–7]. Due to the various definitions of flood risk that are widely acknowledged by many scholars, first we must define flood risk, flood hazard, and social vulnerability for this study. According to the definition from the United Nations Office for Disaster Risk Reduction (UNISDR) Terminology on Disaster Risk Reduction [8], we consider "flood risk" as "the potential flood disaster which leads to losses in lives and property (assets), and also affects the socio-economic development"; "flood hazard" as "a potentially damaging flood event (flood characteristics) that may cause the loss of life or injury, property damage, social and economic disruption"; and "social vulnerability" as "people and property (assets) exposed to the flood hazard". For this reason, flood

risk assessment in this study was evaluated based on flood hazard and social vulnerability data to develop the efficient flood risk reduction measures for CPRB. To lessen the vulnerability of people to flood disasters, the most used strategy is evacuation. Evacuation not only reduces the loss of lives of people, but also helps residents quickly regain their functionality [9].

Floods in the middle CPRB are triggered by long-term rainfalls (May to October) and storm rainfalls (August to October), that cause frequent flooding in low-lying areas and prolonged floods cause human and economic losses. Unfortunately, there is no detailed flood risk assessment in the middle CPRB. Moreover, one of the factors that aggravated the 2011 flood situations in the CPRB was the confusing and contradicting information from key agencies especially the evacuation information. Comprehensively planned evacuations and under-equipped evacuation shelters may cause loss of life due to flooding. Therefore, we investigated the flood risk and developed the emergency flood evacuation model for the middle CPRB.

Basically, most flood evacuations are planned and considered based on a static factor of the flood event, but floods are dynamic processes. Besides, flood evacuation plans need to consider various flood events. To obtain the optimal emergency flood evacuation, we considered various devastating and catastrophic flood events in the CPRB, which occurred during the 1995, 2006, and 2011 flood events. The objectives of this study were to: (1) Establish the flood risk maps during the 1995, 2006, and 2011 flood events considered as the product of flood hazard and social vulnerability maps generated by fuzzy Analytic Hierarchy Process (AHP) and fuzzy logic; (2) develop the emergency flood evacuation model to lessen the vulnerability of citizens and determine the available time for safe evacuation in the middle CPRB; and (3) propose recommendations to improve the efficiency of flood evacuations and reduce the consequences of a catastrophic flood event.

## 2. Materials and Methods

### 2.1. The Study Area

The Chao Phraya River Basin (CPRB); the largest and most agriculturally productive basin in Thailand, has experienced and suffered from flooding almost every year. The CPRB covers an area of 160,000 km$^2$ from northern Thailand to the gulf of Thailand. The basin is normally divided into three basins as shown inthe upper CPRB, and there are four major tributaries; the Ping, Wang, Yom, and Nan Rivers, which flow downstream from the northern mountainous terrains and combine to form the Chao Phraya River at the middle CPRB around the Nakhon Sawan Province.

In Thailand, people usually live in floodplains because land is productive for agricultural purposes. Due to steep slopes and dense forests (60% of areas) in the upper CPRB, the middle basin usually receives a large amount of water that causes flooding in low-lying areas located along the Yom and Nan Rivers almost every year. Water in the channel of Yom and Nan Rivers usually spills over the riverbank or embankment to the floodplains due to mild channel slope and low capacity of rivers. According to the spatial distribution of high rainfall intensity data [10], it clearly shows that the low-lying area in the middle CPRB is relatively high-risk to flooding, which is predominated by high rainfall intensity.

In accordance with the 2011 flood, these low-lying areas suffered and struggled from this catastrophic flood, lasting from the late of July to November. During this time, at least 100 persons died due to this severe flooding [11]. Moreover, it appears that there were no provisions, or plans for large-scale disasters such as the 2011 flood in CPRB [12]. Once floods reached a large extent, the public and many stakeholderscomplained about often receiving conflicting flood information that worsened situations. The limited flood evacuation information, together with the lack of proactive evacuation planning may increase flood-related damages and losses.

The above information upholds the need for a realistic, hands-on evacuation preparation for the middle CPRB, particularly in low-lying flood prone areas. Thus, the flood risk assessment and emergency flood evacuation model were performed and tested in the low-lying areas of the middle CPRB. The study covers an area of 6012 km$^2$ as shown in Figure 1.

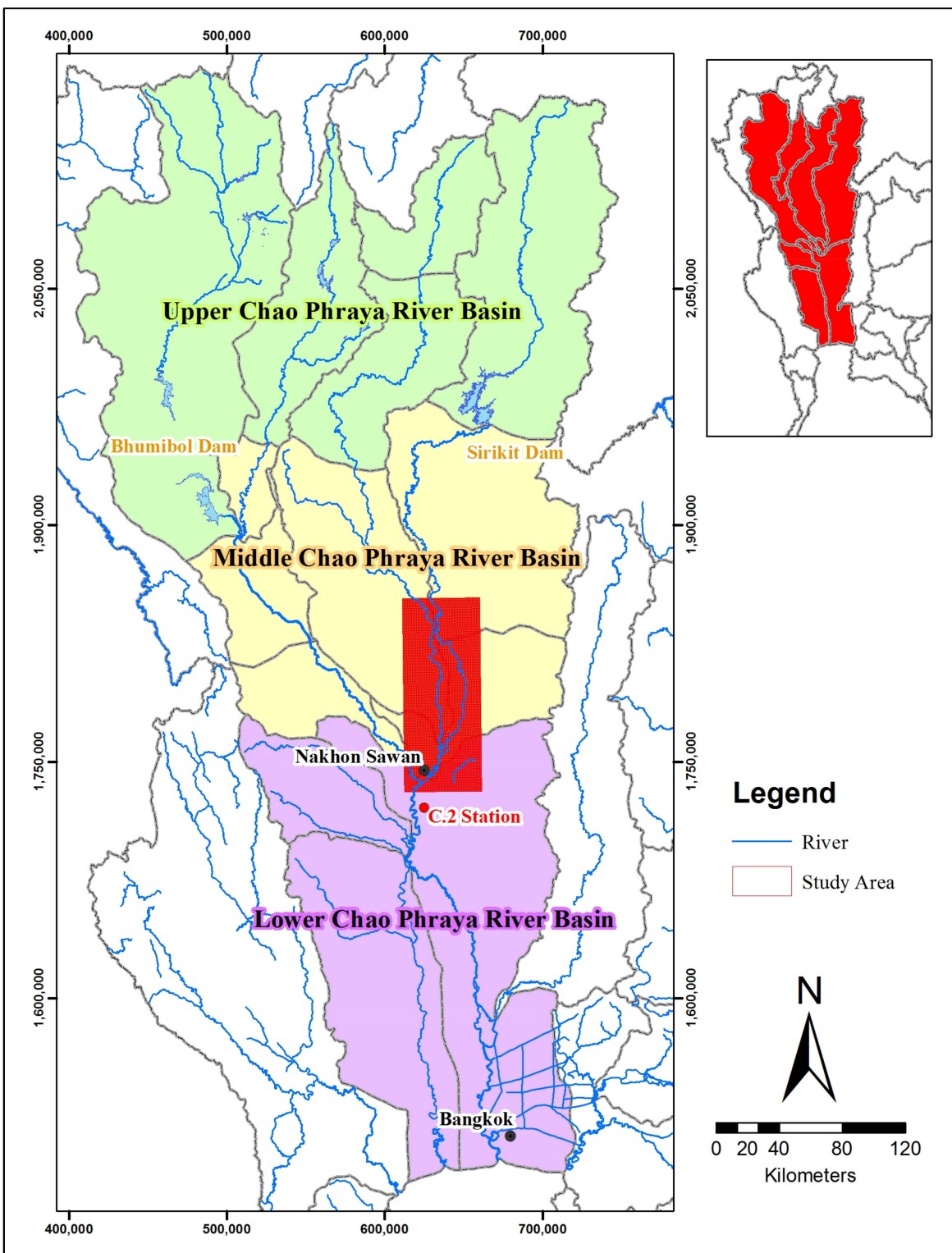

**Figure 1.** Study area for flood risk assessment and the emergency flood evacuation model in the middle reach of Chao Phraya River Basin.

## 2.2. Flood Risk Assessment

In order to understand flood risk, it is necessary to set appropriate conditions of flood events and socio-economic situations. This would permit assessment of floodplains conditions to obtain basic information for considering methods to reduce and avoid risks, and eventually for developing

appropriate adaptation measures [13–15]. The identification of flood risk factors is the most important step of flood risk assessment. The inclusion of flood risk factors should be performed within a framework to ensure that the whole problem is enclosed. It is well known that maps can represent the spatial flood risk distribution, which are easy to understand and provide a stronger impression [16,17]. In this study, flood risk assessment was represented in the form of a flood risk map that was evaluated by two crucial maps, which are flood hazard and social vulnerability maps. We selected historical catastrophic flood events, which occurred during 1995, 2006, and 2011 due to their impacts on vast inundation areas, and the resulting huge economic losses in the CPRB. In addition, we collected secondary data provided by several agencies in Thailand to represent the socio-economic conditions of low-lying areas in the middle reach of CPRB.

Based on the hydrodynamic model (2-D) for the middle CPRB [18], we carried out the simulation results of severe flood events (1995, 2006 and 2011 flood) to get the flood inundation depth, flood velocity, and inundation duration represented in a grid (50 m × 50 m resolution) at different times. Even though floodplain elevation data (30 m × 30 m resolution) was applied for the hydrodynamic model, the results of flood inundation were represented by the 50 m × 50 m resolution due to hardware resources and stability of the model. We counted out these indicators to establish the flood hazard map. In addition, based on the available data and an extensive literature review [19–21], we collected and employed seven indicators in socio-economics to generate the social vulnerability map as shown in Table 1. Through standardized processing, all seven data layers in social vulnerability were converted into grid data.

**Table 1.** Factors for the flood hazard and social vulnerability maps.

| Indicators | Description | Source |
|---|---|---|
| **Flood Hazard Map** | | |
| F1 | Flood Inundation Depth | Jamrussri and Toda (2017) |
| F2 | Flood Velocity | Jamrussri and Toda (2017) |
| F3 | Inundation Duration | Jamrussri and Toda (2017) |
| **Social Vulnerability Map** | | |
| S1 | Census Population | Department of Provincial Administration (2016) |
| S2 | Population Density | Department of Provincial Administration (2016) |
| S3 | Age (lower 6 and upper 60) | Department of Provincial Administration (2016) |
| S4 | Census Housing | Department of Provincial Administration (2016) |
| S5 | Distance to State Highway | Department of Highways (2016) |
| S6 | Land Use | Land Development Department (2009) |
| S7 | Land Price | The Treasury Department (2015) |

Flood risk assessment in this study involves various data. It is unavoidable that the geographic and statistical information used in this study have multiplicity, complexity, and uncertainty [22]. In recent years, several uncertainty methods have been employed to overcome these problems, such as the fuzzy logic method [23]. The fuzzy logic method is usually used in the complex uncertainty problem such as flood risk assessment, because it is simple to describe fuzzy characters and it can also reflect the actual situation on objectiveness [24–27]. For these reasons, we employed the fuzzy logic to assess the flood risk map in low-lying areas of the middle CPRB. Moreover, analytic hierarchy process (AHP) was also applied to estimate weighted indicators. However, the conventional AHP requires crisp numbers that cannot deal with uncertainty like flood risk assessment. Besides, we applied the fuzzy AHP alongside a triangular fuzzy in the pairwise comparison to estimate weighted indicators. The flowchart for flood risk assessment is shown in Figure 2.

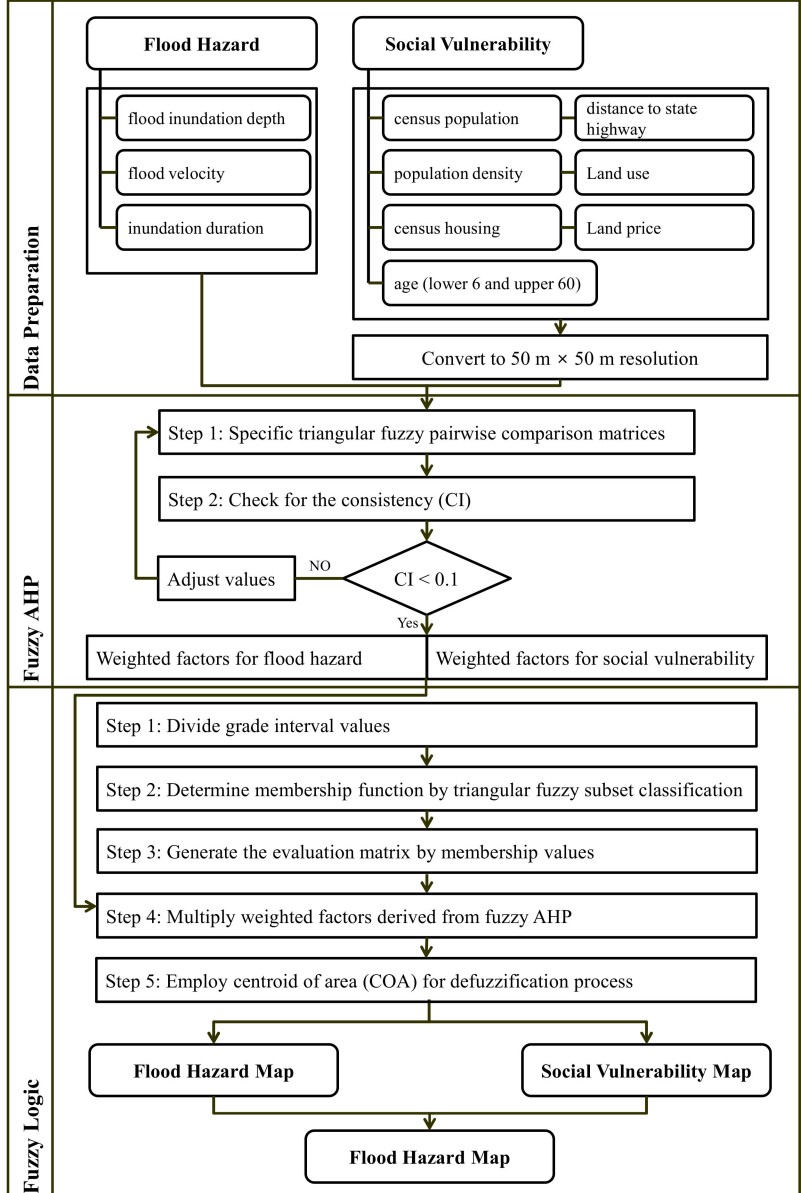

**Figure 2.** Flowchart for the flood risk assessment.

### 2.2.1. Fuzzy Logic

With the advancement of fuzzy logic, some mathematical models have been developed based on fuzzy theories to achieve a greater accuracy. As we mentioned, this study involves many indices, and it is difficult to specify an exact value to them (crisp value), but the fuzzy model provide a possibility to identify their values between 0 and 1. When the value belongs in full membership, it represents 1. In contrast, it will become 0 when that value is a non-membership. Using different types of fuzzy numbers depends on the data and problem [28]. The fuzzy synthetic evaluation (FSE) categorizes data into several groups, and calculates each individual value considering the whole situation [29]. We then applied FSE to generate the flood risk map. In the FSE, the triangular fuzzy numbers were also utilized to express their relative membership. Figure 3 represents the triangular fuzzy numbers and it can be described as:

$$\mu_{\widetilde{N}}(x) = \begin{cases} \frac{x-l}{m-l}, & l \le x \le m \\ \frac{u-x}{u-m}, & m \le x \le u \\ 0, & otherwise \end{cases} \tag{1}$$

where the parameters, *l*, *m*, and *u* are the smallest possible value, the most promising value, and the largest possible value, respectively. The triangular fuzzy numbers are generally represented as (*l*, *m*, *u*).

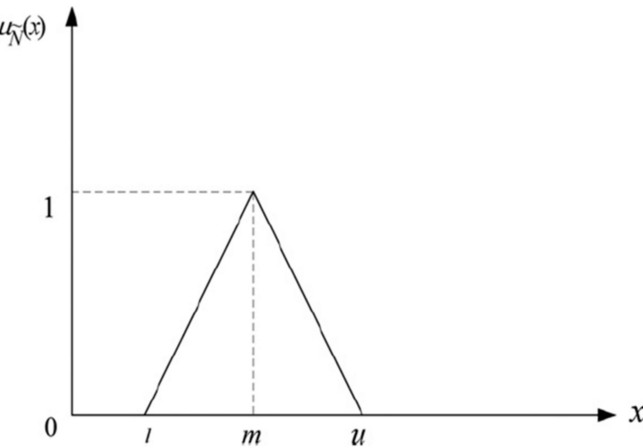

**Figure 3.** Triangular fuzzy members.

### 2.2.2. Fuzzy Analytic Hierarchy Process (Fuzzy AHP)

The Analytic Hierarchy Process (AHP) has been widely used for estimating weight factors in several areas of human interests [30–32]. In the AHP, the weights of criteria are not obviously distinguished, as in the direct assessment of multiple-criteria decision-making (MCDM) methods. The weights are generated from judgement matrices of pairwise comparisons considered by the importance of the criteria proposed by Saaty [33]. As the traditional AHP, the pairwise comparisons are represented as crisp values. However, uncertainties, ambiguities, and vagueness in the real situation cannot be handled by a crisp value. As a result, fuzzy AHP was developed to solve these problems. In this study, we applied the extent study on the fuzzy AHP developed by Chang [34]. In addition, using triangular fuzzy numbers in fuzzy AHP has proven that it is effective for problem statement, where limited data are subjective and ambiguous [35–37]. Therefore, the fuzzy AHP model with triangular fuzzy numbers in a pairwise comparison process was employed to generate weights factor for flood hazard and social vulnerability map in this study.

### 2.3. Emergency Flood Evacuation Model

According to numerous studies on natural disasters, researches primarily highlight the functions of infrastructures and residents responses during extreme event [38]. Evacuation, one of emergency responses, has received significant attention to be utilized to lessen the vulnerability of threatened people [39]. Many studies identified a number of influential factors considering a complexity of social and engineering sciences. However, investigation of these factors depends largelyon the available data, and results on these effects vary from notable to unnotable across types of disasters [40]. Many researches on evacuation tried to develop a model to estimate the travel time of evacuation, and determine the appropriate evacuation routes. However, most of these models focus on hurricanes in developed countries [41–45]. Despite this acknowledgement, understanding and developing a flood evacuation model is appropriate and necessary [46].

In Thailand, many field investigations often reported that many people lost the chance to evacuate due to their late decision and limited information [12]. Due to the vast flood inundation and general well-being of communities in the middle CPRB, we identified the high-risk areas in this basin to be the pilot study for flood evacuation. Thus, we developed a mathematical model, namely the emergency flood evacuation model, to evacuate citizens in this area to flood shelters in order to facilitate an evacuation plan and determine the available time for evacuation as a protective action strategy.

The three-step approach proposed in this study was adopted for the emergency flood evacuation in low-lying areas of the middle CPRB. For the first step, we investigated flood characteristics of

the 1995, 2006 and 2011 floods, and then classified flood evacuation zones to specify evacuees and the starting time for evacuation. In the next step, the designed safe areas for flood shelters were determined in geographical information systems (GIS) using topographic maps at a scale of 1:50,000. In the last step, the coordinates of high-risk grid cells, state roads, and flood shelters as well as the distance among these three layers were extracted from GIS to be used as input data to calculate the evacuation travel time. The evacuation travel time in this model was calculated based on the physical status of evacuees (elderly and preschool citizens), safe evacuation condition, the shortest time of evacuation, flood shelter, and road capacity.

## 3. Results

### 3.1. Flood Risk Maps for Middle Reach of CPRB

Flood risk maps in low-lying areas of the middle CPRB were developed using flood hazard and social vulnerability maps through fuzzy AHP and fuzzy logic. The flood risk assessment model (Figure 2) was set up to generate flood risk maps. The specific technique of this model is described as follows:

1.  According to the basic assessment units, we counted out three flood hazard indicators, flood inundation depth, flood velocity, and inundation duration as Table 1. On the other hand, social vulnerability assessment, there is a corresponding seven-indicator as Table 1, consisting of census population, population density, age (lower 6 and upper 60 years old), census housing, distance to state roads, land use, and land price.

2.  We collected and converted the socio-economic data from various agencies to grid data with a resolution of 50 m × 50 m resolution corresponding to hydrodynamic model simulation results.

3.  For the fuzzy AHP model, we specified triangular fuzzy pairwise comparison matrices. Pairwise comparison matrices were formed using characteristics of the middle CPRB and literature reviews [47,48], which were further analyzed and formed in detail by the expert teams in Japan and Thailand. The local weights of these two factors for flood risk maps were determined by the fuzzy scale regarding relative importance to measure the relative weights as given in Table 2. After analyzing, the local weights of flood hazard and social vulnerability map that passed the consistency test are presented in Table 3. For the social vulnerability map, the population density and land price were selected as main indicators because they reflected the economic condition. Floods occurring in developed areas always cause more economic loss than those in developing and undeveloped areas.

4.  According to basin characteristics, historical information, literature reviews, and so on [49–51], we divided the grade interval value for the flood risk map, flood hazard map, and social vulnerability map into five grades; noted as low zone, low-medium zone, medium zone, medium-high zone, and high zone as shown in Table 3.

5.  Through the piecewise linear function (triangle) in the fuzzy logic, the membership function of each grade was calculated and eventually the assessment factors were obtained by the fuzzy subset classification (Figure 3). The parameters $l$, $m$, and $u$ in this study were the lowest value, the middle value, and the highest value of each grade interval value, respectively (Table 3).

6.  The evaluation matrices were generated by the membership values and were then multiplied by weighted factors derived from the fuzzy AHP. The defuzzification was the last step in the fuzzy logic process. We applied centroid of area for the defuzzification process and eventually the flood hazard and social vulnerability assessment results can be obtained through fuzzy logic. The flood hazard and social vulnerability maps are shown in Figure 4.

7.  With the flood hazard and social vulnerability results, eventually we were able to generate the flood risk maps for low-lying areas in the middle CPRB through fuzzy logic as illustrated in Figure 5, and the flood risk results are summarized in Table 4.

**Table 2.** The linguistic fuzzy scales for importance of flood hazard and social vulnerability factors.

| Linguistic Scale for Importance | Triangular Fuzzy Scale |
|---|---|
| Just Equal | (1, 1, 1) |
| Equally Important | (1/2, 1, 3/2) |
| Weakly More Important | (1, 3/2, 2) |
| Strongly More Important | (3/2, 2, 5/2) |
| Very Strongly More Important | (2, 5/2, 3) |
| Absolutely More Important | (5/2, 3, 7/2) |

**Table 3.** The factors classification of the flood hazard map, social vulnerability map, and flood risk map including the triangular fuzzy members (*l*, *m*, and *u* in Figure 3).

| Flood Hazard Map | Grade Interval Value | | | | |
|---|---|---|---|---|---|
| Factor type | Weight factor | Low hazard zone (*l-u*) | Low-medium hazard zone (*l-u*) | Medium hazard zone (*l-u*) | Medium-high hazard zone (*l-u*) | High hazard zone (*l-u*) |
| F1: Flood Inundation Depth (m) | 0.429 | 0–0.52 | 0.30–1.12 | 0.52–1.75 | 1.12–2.25 | 1.75–18.50 |
| F2: Flood Velocity (m/s) | 0.206 | 0–0.30 | 0.10–0.75 | 0.30–1.25 | 0.75–2.00 | 1.25–7.80 |
| F3: Inundation Duration (days) | 0.364 | 0–4 | 1–11 | 4–22 | 11–45 | 22–61 |
| **Social Vulnerability Map** | **Grade Interval Value** | | | | |
| Factor type | Weight factor | Low vulnerability zone (*l-u*) | Low-medium vulnerability zone (*l-u*) | Medium vulnerability zone (*l-u*) | Medium-high vulnerability zone (*l-u*) | High vulnerability zone (*l-u*) |
| S1: Census Population (population) | 0.109 | 0–15 | 10–25 | 20–40 | 30–100 | 75–850 |
| S2: Population Density (population/km²) | 0.204 | 0–63 | 40–113 | 63–225 | 113–1000 | 225–3500 |
| S3: Age (lower 6 and upper 60) (population) | 0.136 | 0–2 | 1–5 | 3–10 | 8–15 | 12–150 |
| S4: Census Housing (housing) | 0.111 | 0–5 | 2–10 | 7–20 | 15–40 | 30–350 |
| S5: Distance to state highway (km) | 0.116 | 0–5 | 2.0–9.5 | 5.0–14.5 | 9.5–18.5 | 14.5–21 |
| S6: Land Use (type) | 0.113 | 0–1.5 | 1–2.5 | 1.5–3.5 | 2.5–4.5 | 3.5–5 |
| S7: Land Price (Baht/1600 m²) | 0.212 | 0– 35,000 | 18,000–75,000 | 35,000–550,000 | 75,000–3,750,000 | 550,000–7,000,000 |
| **Flood Risk Map** | **Grade Interval Value** | | | | |
| Factor type | Weight factor | Low risk zone (*l-u*) | Low-medium risk zone (*l-u*) | Medium risk zone (*l-u*) | Medium-high risk zone (*l-u*) | High risk zone (*l-u*) |
| Flood Hazard Map | 0.50 | 0–1.5 | 1.0–2.5 | 2.5–3.5 | 3.0–4.5 | 4–5 |
| Social Vulnerability Map | 0.50 | 0–1.5 | 1.0–2.5 | 2.5–3.5 | 3.0–4.5 | 4–5 |

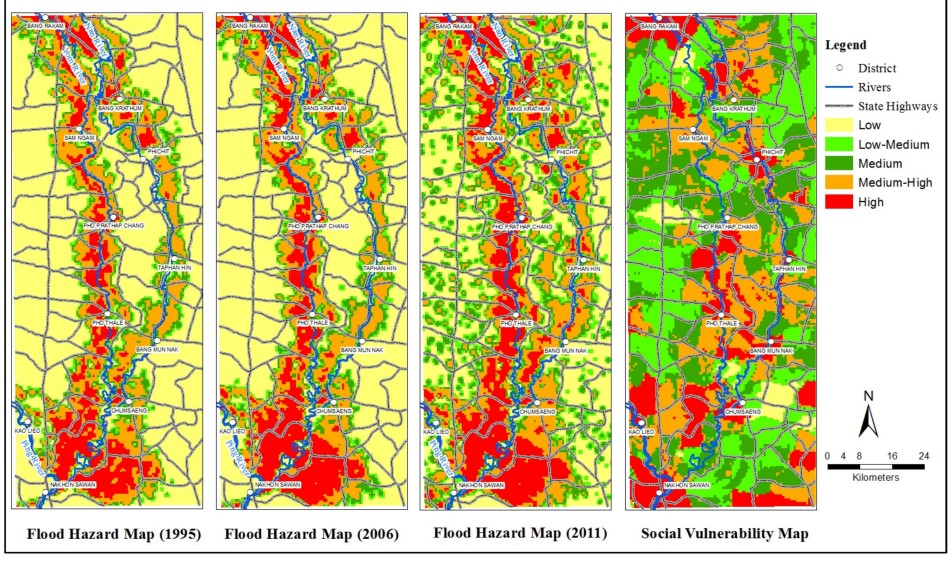

**Figure 4.** Flood hazard and social vulnerability maps for low-lying areas in the middle CPRB.

**Table 4.** Percentage of risk zones of the 1995, 2006, and 2011 floods in low-lying areas of middle CPRB.

| Flood Events | Low Risk Zone | Low-Medium Risk Zone | Medium Risk Zone | Medium-High Risk Zone | High Risk Zone |
|---|---|---|---|---|---|
| 1995 Flood | 8.25% | 41.02% | 27.53% | 19.06% | 4.14% |
| 2006 Flood | 8.05% | 39.66% | 26.31% | 20.84% | 5.14% |
| 2011 Flood | 2.87% | 30.62% | 33.87% | 26.25% | 6.39% |

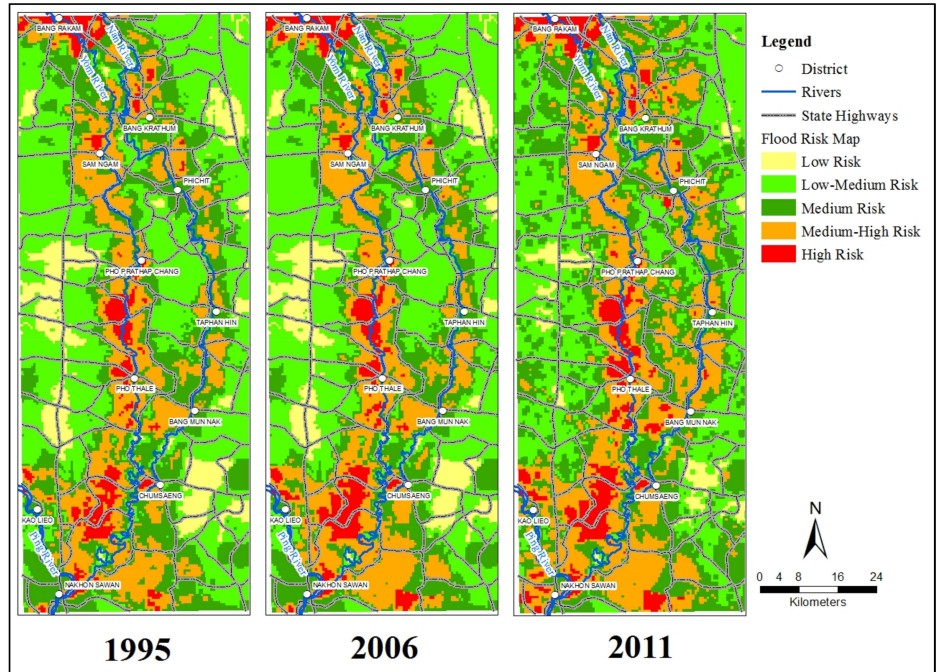

**Figure 5.** Flood risk maps for low-lying areas in the middle CPRB in 1995, 2006 and 2011.

Flood risk maps enable decision makers to clearly identify risk areas. Comparing components of risk in quantitative terms is the one of advantages of such a comprehensive risk assessment. In this study, we examined the flood risk as a result of flood hazard and social vulnerability maps. For flood hazard maps (Figure 4), most of the high flood hazard areas lay along the Yom River and span around the confluence of the Chao Phraya River owing to the mild slope. On the other hand, the social vulnerability map showed that the high social vulnerability areas were located in urban areas because of the well-being of communities and the dense population (Figure 4). With the results of flood hazard and social vulnerability maps, the relative flood risk maps for each 50 m × 50 m grid resolution in this study showed that almost all low and low-medium risk zones lay in the agricultural areas especially a paddy field. As the study was used for agriculture purpose (low impact on social vulnerability) and was not inundated, the low and low-medium risk zones occupied almost 50% of the study area. On the other hand, it could obviously be seen that the medium-high and high flood risk areas spanned along the rivers (mostly inundated), especially at the Yom River. As the floodplain areas are fertile and flat, which is suitable for various purposes. Besides, many urban areas are located in floodplains. The large spread of high flood risk areas is located in the Bang Rakam, Pho Thale, Chum Saeng, and Nakhon Sawan District (Figure 5), which correspond to the report from the Royal Irrigation Department (RID) indicating that these areas are ubiquitous and severest in this basin owing to flooding [52].

According to Figure 5, the common high flood risk area which appeared in all three severe flood events is around 96 percent compared to the 1995 flood; the smallest flood event in this study. It could be stated that these areas are frequent to high-risk and there are more than 42,000 citizens who live and spend their daily lives in these high-risk areas. It is very important and necessary for relevant agencies to prepare and facilitate the effective disaster planning and management. As the results of the 2011

flood, it was notable that relying only on structural measures is not a proper way for the CPRB. As the community areas are adjacent to rivers, in low-lying areas of the middle CPRB, it is not possible to avoid high material damage when an extreme flood arrives, but it is possible to save lives of residents. Many countries use the lessons learned post-disaster to revise their disaster management legislation and plans. Therefore, we intend to develop the emergency flood evacuation model for devastating flood events to lessen the risk of people especially those living in high flood risk areas and explore the safety evacuation with a different starting time of evacuation.

*3.2. Emergency Flood Evacuation*

According to high flood hazard and community welfare, we focused on the high flood risk areas of the middle CPRB as the pilot area to evacuate residents to flood shelters. Owing to wide-area evacuation in this study, we identified the flood evacuation zone by considering various flood characteristics, and also selected existing public buildings to be flood shelters. Moreover, we developed the emergency flood evacuation model to investigate the safe evacuation and available time for evacuation in the middle CPRB. The flood evacuation model for this study determined the travel time of the evacuees from their houses to flood shelters. The travel time of evacuation depended very much on the flood situation such as flood depth, flood velocity, and flood extents. Besides, the evacuation travel time was further divided into two categories; travel time of walking and vehicle, and these travel times were examined by factors that affected evacuation behaviors. The concept of the flood evacuation model from this study was that all evacuees who lived in the same flood zone evacuated at the same time of flood warning announcement, and complied with instruction. Furthermore, this study also calculated the success of evacuation that reflected the late decision on flood evacuation from relevant agencyies, because some evacuees cannot move to flood shelter with safe. Hopefully, the results from this study can reduce the loss of life during flood evacuation in the middle CPRB in which flooding occurs almost every year.

3.2.1. Flood Evacuation Zones Classification

With wide-area evacuation in this study, firstly we defined the flood evacuation zones. The evacuation zone is very important because if it is not conducted and planned effectively, the mass evacuation can cause traffic congestion and also leave evacuees in dangerous consequences resulting in unexpected losses. Wilmot and Meduri [53] stated that evacuation zones should be conducted based on an expected event with traffic analysis and the number of zones should be minimized for easy and fast communication.

An understanding in flood mechanism and timing is a very crucial step to develop flood emergency plans effectively because it avoids problems that move residents to subsequently inundated areas. Besides, we combined spatial and temporal flood inundation results of the catastrophic flood events in 1995, 2006, and 2011 from the hydrodynamic model together with flood risk maps to classify and specify flood evacuation zones. Identifying evacuation zones in this study gave a priority on resident similarities, such as residential district or neighborhood association because it is effective to facilitate mutual cooperation among residents during evacuation. After analysis, we found that although there was a slight difference in arrival time of flood water owing to the different shape and amount of flood hydrograph, there was still a significant common pattern of flood inundation in the study area. This may have come from a low-lying area and a very large scale flood event that caused water instantly and easily to spill over the river bank into floodplains. According to the classification results that considered an overall similarity, we could account for five flood evacuation zones and we also could specify a starting evacuation time for each zone considered by the arrival time of flood-water coming into each zone. The flood evacuation zones map for the middle CPRB is presented in Figure 6, and the detail of each flood evacuation zone against flood events is summarized in Table 5.

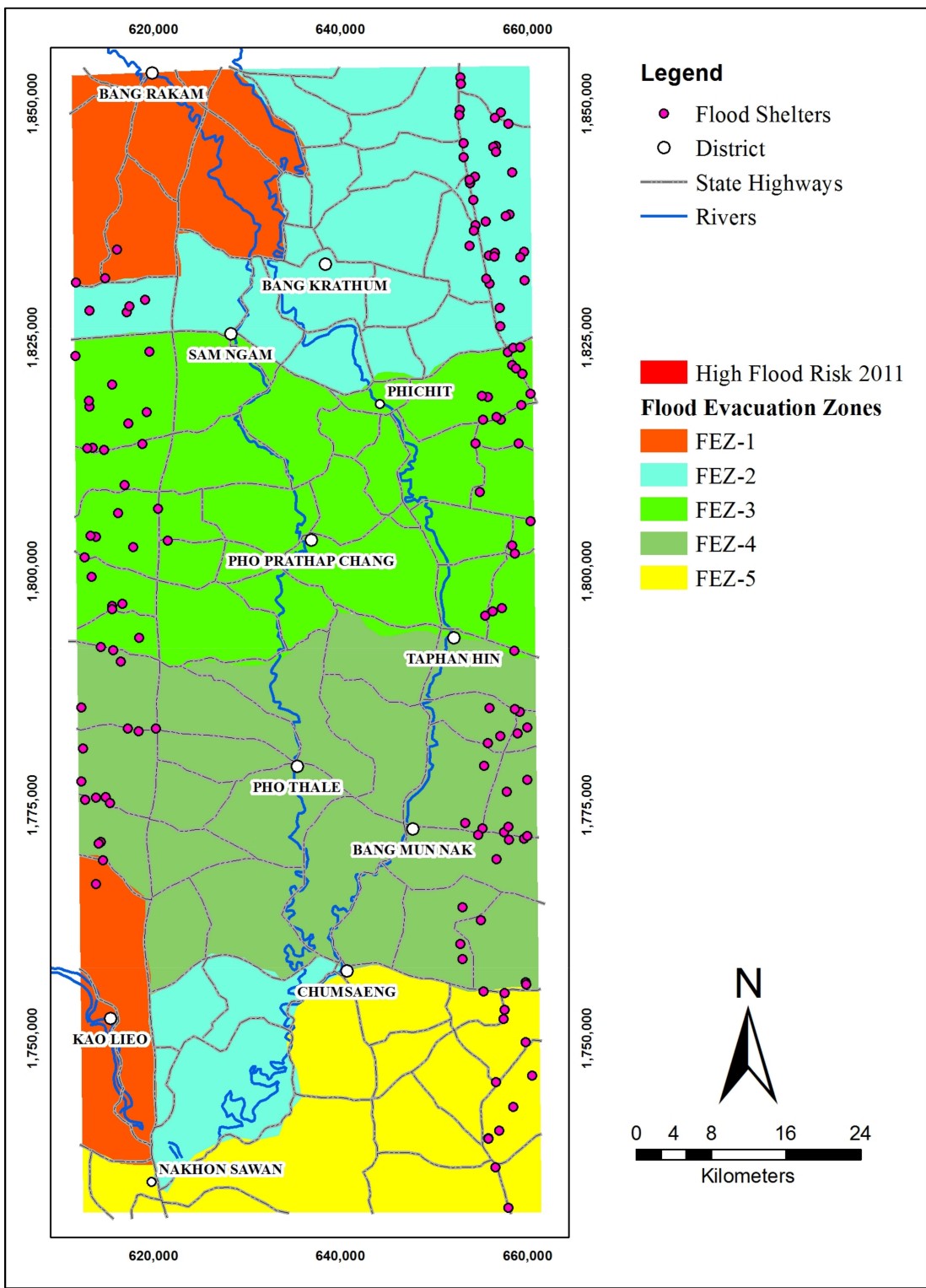

**Figure 6.** Flood evacuation zones for the middle CPRB.

**Table 5.** The detail of each flood evacuation zone in the middle reach of CPRB.

| Flood Evacuation Zone (FEZ) | FEZ 1 | FEZ 2 | FEZ 3 | FEZ 4 | FEZ 5 | Sum. |
|---|---|---|---|---|---|---|
| **1995 Flood** | | | | | | |
| No. of high flood risk (grid) | 247 | 316 | 160 | 220 | 47 | 990 |
| Evacuees | 14,397 | 9192 | 6802 | 8938 | 4395 | 43,724 |
| Elderly and preschool (%) | 10.20 | 13.79 | 15.57 | 15.52 | 13.04 | 13.39 |
| Vehicle in use (private cars) | 3600 | 2298 | 1701 | 2235 | 1099 | 10,933 |
| Starting evacuation time (day) | 1 | 2 | 4 | 5 | 7 | - |
| **2006 Flood** | | | | | | |
| No. of high flood risk (grid) | 312 | 393 | 170 | 263 | 90 | 1228 |
| Evacuees | 17,129 | 10,305 | 7138 | 10,097 | 5500 | 50,169 |
| Elderly and Preschool (%) | 10.47 | 16.23 | 15.11 | 16.96 | 16.98 | 14.33 |
| Vehicle in use (private cars) | 4283 | 2577 | 1785 | 2525 | 1375 | 12,545 |
| Starting evacuation time (day) | 1 | 2 | 4 | 5 | 7 | - |
| **2011 Flood** | | | | | | |
| No. of high flood risk (grid) | 349 | 418 | 221 | 400 | 140 | 1528 |
| Evacuees | 18,826 | 12,963 | 10,283 | 14,383 | 19,958 | 76,413 |
| Elderly and Preschool (%) | 10.17 | 14.41 | 12.36 | 19.93 | 6.22 | 11.99 |
| Vehicle in use (private cars) | 4707 | 3241 | 2571 | 3596 | 4990 | 19,105 |
| Starting evacuation time (day) | 1 | 2 | 4 | 5 | 7 | - |

### 3.2.2. Flood Shelters Selection

The shelters should have adequate space, basic living requirements, and not be located in hazardous areas [41]. GIS have widely been applied to identify and select flood shelters [54–57]. In this study, we aimed to find appropriately located flood shelters. Firstly, we sought for candidate shelter locations by considering existing public buildings in this study area such as schools, temples, and community centers that had the adequate area and necessary utilities for the number of evacuees to stay and store relief packages. The candidate flood shelters were considered by GIS through topographic maps at a scale of 1:50,000. In accordance with the 2011 flood event, many flood shelters provided by agencies were located in inundated areas and some of them had to be closed owing to the shelters being submerged, causing trouble to citizens. Thus, flood shelters in this study were set to be free from flood inundation records. We then applied the vastest flood inundation area (flood 2011) to screen flood shelters.

After overlaying the 2011 flood inundation with topographic maps, we obtained 142 buildings located in safe areas from flooding as shown in Figure 6. Due to the unknown capacity of each flood shelter, we assigned its capacity as 500 persons, which is the average flood shelters capacity reported by Department of Disaster Prevention and Mitigation [58]. Besides, the total capacity of flood shelters in this study was 71,000 vacancies.

### 3.2.3. Evacuation Travel Time Calculation

In accordance with the 2011 flood, many transportation infrastructures, 1700 roads, highways, and bridges, were damaged and the economic cost of these infrastructures was around US $4.5 billion [59]. Besides, we assumed that only state roads were not inundated and available for a transportation network during flood evacuation. Another assumption was that all evacuees in each grid cell evacuated together and complied with the orders of evacuation. In this study, the evacuation travel time calculation was classified into three stages; (1) the travel time from high flood risk to state roads (walking travel time), (2) the travel time along roads network (vehicle travel time), and (3) the travel time from state roads network to flood shelter (walking travel time).

### 3.2.4. Travel Time of Walking

The walking time, an important parameter used in evacuation models, is the primary stage of access to flood shelters when there is no access road. We divided the travel time of walking into two phases; (1) from high flood risk grid cell to a state road, and (2) from a state road network to flood shelters. Walking speed usually varied with many factors, such as walking types, walking conditions,

occupant types, and place types [60–63]. A high proportion of elderly and preschool people in the residents also influenced the evacuation responses [64–66]. Elderly and preschool evacuees walk slower than adults due to their weak physical status and level of prompting. According to the study on walking speeds [67], we assumed the walking speed for a normal evacuee (adult group) as 1.4 m/s. Owing to that, we assumed that all evacuees in each grid cell evacuated together, the average walking speed in each grid cell was calculated by the percentage of elderly and preschool evacuees belonging to each grid cell. It could be stated that the greater proportion of elderly and preschool evacuees, the more walking travel time will be taken.

Considering severe flood events in this study, it is extremely dangerous for evacuees to walk across inundated areas. Therefore, it is necessary to assess the certain degree of safety for evacuees walking through floodwaters [68]. Estimating loss of life in flood disaster were widely examined [69,70], but life-threatening conditions during flood evacuation have rarely been investigated [71–73]. Ishigaki et al. [72] proposed a safety condition of specific force per unit width along the evacuation route expressed as:

$$\frac{u^2 h}{g} + \frac{h^2}{2} < 0.125 \qquad \left(\text{m}^3/\text{m}\right) \tag{2}$$

where $h$ (m) is water depth, $u$ (m/s) is flow velocity, $g$ (m/s$^2$) is the acceleration due to gravity, 0.125 (m$^3$/m) is the threshold value for safe evacuation by walking during flood.

Besides, we then adopted this parameter, safe evacuation condition, along with the results from the hydrodynamic model to determine in grid cell is safe for evacuation. We assumed that during evacuation, evacuees can not walk across to another grid cell beyond the threshold value for safe evacuation conditions; 0.125 m$^3$/m. It could be stated that if the eight points surrounding the considered grid cell have a safety condition value greater than 0.125 m$^3$/m, then evacuees in that grid cell can not evacuate to flood shelters because there I a chance that they will lose their lives. To reduce the complexity of modeling, we calculated safe evacuation conditions on an hourly basis. After set up assumptions and constraints, the next task was to exact the coordinates among high flood risk grid cells, state roads network, and flood shelters to calculate the distance among them. Additionally, the coordinates of the centroid of high flood risk grid cells were considered as their location. Eventually, the optimum travel time of walking was determined by choosing the shortest path satisfying safe evacuation constraint, and also by the proportion of elderly and preschool evacuees in the grid cell.

### 3.2.5. Travel Time of Vehicles

Macroscopic and microscopic models are usually represented as evacuation models. A macroscopic model represents traffic as a flow. This model generally focuses on the evacuation travel time and is used for wide-area evacuation. The microscopic model examines traffic and vehicles on more circumstantial levels. Due to the wide-area evacuation, and considering only the state roads network, we formulated the emergency flood evacuation model as macroscopic based on a static traffic model. The static traffic model is normally used to evaluate current and future use of road networks. Traffic flow in a static traffic model is assumed to be constant from the origin to final destination. The free-flow (base) speed represents the average speed of vehicles that are not constrained by any disruption such as traffic control and roundabout. According to the study on free-flow speed during hurricane evacuations from Dixit and Wolshon [74], we then modified these values considering the roads capacity in the study area. Eventually we carried out a free-speed for state roads assumed as 40 km/h. Moreover, Highway Capacity Manual (HCM) [75] defines road capacity as the maximum hourly flow rate of expected vehicles passing a point during a specified time period. Department of Highways [76] reported that the average state roads capacity around our study area is 1897 vehicles per hour. We then brought this value into consideration as one of constraints that limited outbound flow rate during evacuation through the roads network in our emergency flood evacuation model.

With the study area covering approximately 6012 km$^2$ and state roads network including over 1545 km of roads (Figure 6), the paths between high flood risk areas and flood shelters using state roads

as a transportation network were determined by a GIS network analysis tool. A road network database on state roads in the study area was obtained from the Department of Highways. Furthermore, we assumed a private car was the primary vehicle to bring evacuees to flood shelters and vehicle occupancy was set as four evacuees per vehicle. Consequently, the travel time of each vehicle was then calculated, where each vehicle was routed to the nearest flood shelter under the shortest path. According to the 500-evacuee flood shelter, it is impossible to accommodate all evacuees who come from the same grid cell in the same flood shelter. Therefore, the emergency flood evacuation model took the residual evacuees to another shelter, which is the second shortest flood shelter.

*3.3. Computational Results of Emergency Flood Evacuation Simulations*

The emergency flood evacuation model was tested and explored the evacuation trip for the middle reach of CPRB as shown in Figure 7. The experimental results were presented in two indicators. The first, the evacuation travel time (Figure 7), described the time it took the evacuees to safely move from their grid cells to flood shelters, where safety wasdetermined by the safe evacuation condition proposed by Ishigaki et al. [72]. The second indicator was the percentage of success evacuation, as shown by the bar chart in Figure 7, describes the percentage of evacuees that could move to flood shelters safely. Once time was limited, the percentage of success evacuation was applied to indicate the loss of life during flood evacuations. Furthermore, both indicators were examined at different staring time choices for evacuation along the horizontal axis as shown in Figure 7. The evacuation travel time and percentage of success evacuation were calculated based on the physical status of evacuees (elderly and preschool citizens), safe evacuation condition, the shortest evacuation path, shelters, and road capacity. As we mentioned, the staring time for evacuation in each flood evacuation zone had a different starting time considered by the arrival time of flood-water coming into each zone. Depending on the characteristics of each zone, such as flood conditions during evacuation, the number of residents, a road network density, and existing flood shelters, the need for complete evacuation can be examined. Moreover, in this study the different time choices for evacuation related to the evacuation warning dissemination in particular were investigated using this model to determine the late evacuation for each zone to execute a suitable evacuation strategy or to develop the leading time of flood forecasting and warning program. In the most severe case, the 2011 flood event, 76,413 evacuee-agents and 19,105 vehicle-agents were treated in an area of 6012 km$^2$.

Based on simulation results, we showed that we could evacuate all residents to flood shelters safely in flood evacuation zones 1, 2, 3, and 4. Unfortunately, in zone 5 we were not able to move all evacuees to flood shelters due to the limited capacity of flood shelters (71,000 vacancies) for all evacuees in the 2011 flood event (76,413 evacuees). Evacuation is a possible risk management measure for flood disasters. When an evacuation begins late, not all evacuees can leave their home to flood shelters safely. Different starting time choices for flood evacuations can be performed to select the suitable strategy under actual circumstances. In zones 1, 2, 3, and 4, we found that the residents had a chance to evacuate safely within 6 h, 6 h, 9 h and 6 h of the evacuation time announcement, respectively. If we neglect the insufficient flood shelters for evacuees in the case of the 2011 flood, we have around 12 h for safe evacuation.

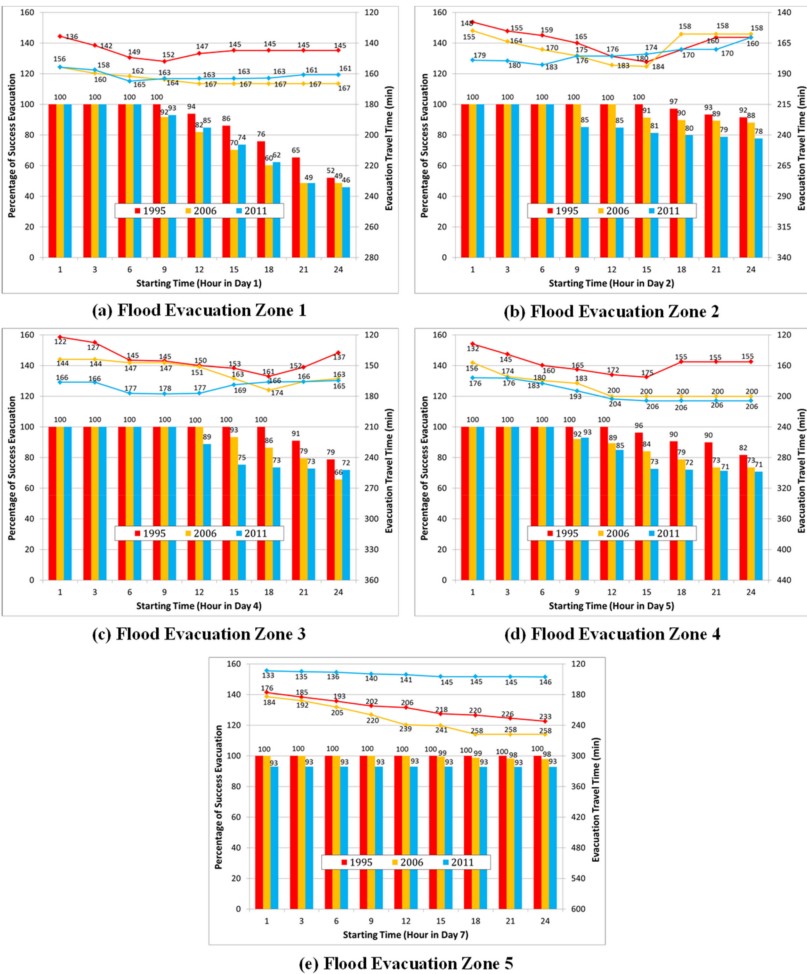

**Figure 7.** Computational results of emergency flood evacuation simulations.

As the study area is low-lying area, the flood inundation gradually occurs and rises from the riverbank, and overland flow from the upstream areas. As the United Nations Development Programme's (UNDP) report [12], even though the flood forecasting and warning system is effective for around 12 h on average, many people have lost the chance to evacuate because they believe this system is not trustworthy and reliable. Therefore, except the insufficient flood shelter provisions from simulation results, it could be mentioned that all residents in the middle CPRB can evacuate to flood shelters safely if they have enough time. More importantly, the available time for safe evacuation shows the need for time to make a decision for evacuation in the middle CPRB. Therefore, main government agencies can adopt these values to execute an evacuation strategy under the required time [77–79]. Furthermore, the required time for evacuation can help relevant authorities create the proper conditions for evacuation such as the contra flow.

Moreover, the results show that the evacuation travel time for the 1995 and 2006 flood events in zone 5 were dominated by flood shelters since the nearest flood shelters were occupied by the previous flood zones. Besides, the location and coverage of flood shelters are some of the important factors for flood evacuation. Due to evacuation during catastrophic flood events, the safety conditions then play a crucial role in flood evacuations that made evacuees walk further to escape unsafe grid cells. For the safe evacuation of all flood events, we also found that the evacuation travel time in the middle CPRB on average took around 170 min to evacuate all residents to a safe place. Moreover, when we compared the walking time to vehicle time in first hour of evacuation to minimize the effect of safe evacuation condition, we found that the walking time took around 70 percent of traveling time. It could be stated that the density of road networks also plays a crucial role in flood evacuation.

## 4. Discussion

The Chao Phraya River Basin (CPRB) has suffered severe flooding causing major damage, fatalities, and economic losses almost every year. Flooding in the CPRB is controlled by storing water within multipurpose reservoirs in the upper CPRB (Figure 1), and excess discharge then flows downstream to the lower basin. In spite of dykes along the rivers in the middle CPRB, floodwater usually spills from the rivers onto the floodplains to cause serious impacts, especially in the Yom Basin. As the CPRB affects flood disaster almost every year, Thai government agencies have implemented flood forecasting and warning systems in this region from the upstream through downstream areas. There are several flood measures proposed for CPRB from previous studies, which mostly reduce one or several hydraulic parameters that characterize a flood pattern such as the volume of runoff, peak discharge, duration and time of rise, velocity, and depth or extent of the inundation area [80–83]. However, to date, no actions have been taken, and no progress has been made towards implementing additional flood countermeasures in the CPRB, except for urgent measures such as dredging to improve drainage and structural rehabilitation.

Structural flood control, although relatively successful to date, is not likely to prevent substantial damage from occurring in CPRB when the next flood strikes as in the 2011 flood. In the meantime, Thailand needs to take decisive action immediately to modify the current reactive, structural approach, into a more proactive non-structural strategy for flood control management and sustainability in CPRB. To more effectively protect residents and communities, a comprehensive flood risk assessment is urgently and criticaly required to strengthen their resilience. Since structural measures alone cannot deal with all disasters, non-structural measures especially flood evacuation plans are particularly important to reduce such damages. Based on results from this study, the important finding was that the government agency should prepare infrastructures, organize evacuation, and make a decision for flood evacuation in the middle CPRB within 6 h as the minimum time when flood water arrives at each zone. To enhance the effectiveness of evacuation, we recommend significant structural and nonstructural measures, which will facilitate the smooth and fast flood evacuation and also can increase the evacuation success rate. These six recommendations address issues, which are pivotal to eradicate the most obvious flood management problems revealed in the 2011 flood. They are as follows:

### 4.1. Structural Measures

#### 4.1.1. Road Network Rehabilitation and Traffic Management

According to the 2011 flood, road networks are severely affected by a long duration of submergence. During the 2011 flood, many local government offices reported that materials support from the District and Provincial offices was difficult to deliver due to transportation interruption by flooding [12]. Therefore, the road networks should be improved and be more susceptible to damage from flood events. Increasing susceptibility of roads to flood damage not only reduces weakened road structures, but also improves traffic loadings. Road improvement works should be considered a more comprehensive and consistent hydraulic design.

Generally, residents will act spontaneously when there is no information and these responses can bring an overload of road systems that may affect the evacuation travel time, resulting in less effective use of infrastructure. Besides, traffic management during flood evacuation should be implemented to make road networks more robust. Traffic management can create the optimal circumstances for evacuees and substantial capacity of evacuation such as the contraflow system [84].

#### 4.1.2. Flood Shelter Provisions

Based on assumptions from this study, flood shelters were only 142 buildings within an area of 6012 km². It has been found that these were not sufficient for the number of victims in the 2011 flood. As flood shelter have effects on evacuation as mentioned in Section 3.3, existing flood shelters should be modified and improved to enhance their capacity. High resolution of topographic maps should be

applied to identify and prepare other appropriate buildings to accommodate mass evacuees as the case of the flood in 2011. Furthermore, new public facilities should be constructed at appropriate locations. Many guidelines on flood shelters against different types of natural disasters are available [85–88].

*4.2. Nonstructural Measures*

4.2.1. Timely and Effective Flood Forecasting and Warning Systems

As shown in Figure 7, the percentage of evacuation success depends very much on the starting time of evacuation. To increase the rate of evacuation success and reduce evacuation travel time, the available time window plays an important factor for flood evacuation. The available time for evacuation can be improved through flood forecasting and warning systems that provide a longer available time and decrease the possibility that residents cannot evacuate to flood shelters related to flood circumstances. Early warning is an important precondition and allows implementation of more emergency measures. One of reasons why private households and businesses did not comply with emergency measures was the lack of time [89]. Many studies stated that the precise and timely dissemination of information provided the better preparedness and reduced loss of life due to flooding [90–96]. Despite the availability of flood forecasting and warning systems to calculate and forecast river and flood inundation in the CPRB, the operational use of such calculations was not effective for flood disaster management actions by the decision makers and government agencies, probably because there was no proper mechanism of systematically utilizing the outcome of such calculations. During and after the flood in 2011, there were outcries for the government to provide more accurate flood information on the broader situations, both in space and time, rather than informing only the present level of flood-water.

4.2.2. Land Use Regulation

Risk awareness remains low among government agencies and people in Thailand [97]. In spite that many safety campaigns have been utilized, most Thai people still believe that their country is safe, and pay less attention to the impact of disaster and the ways to mitigate it. Moreover, policies development especially in land use planning and urbanization were not comprehensively enough to recognize the potential risks. It may in turn cause more vulnerability and increase a chance for man-made disasters. Government agencies should enforce a land use planning policy to detect new urban areas and industrial estates in high flood risk areas of floodplains [98–101]. The new development should also be implemented with adaptive measures in risk reduction [102]. It is found that local governments are usually hesitant to follow the land use regulation but it can lubricate by-law [103]. Furthermore, living with floods through land use planning is also believed to increase public awareness resilience.

4.2.3. Community Participation and Education

Evacuation is one of the integral parts of emergency responses. While there is guidance on evacuation orders and plans in place, there is a pivotal problem that people do not evacuate when they can but they want to evacuate when situation becomes severe. Many Thai people believe that relief and assistance will be delivered to their doorsteps. Such belief comes from their experiences of doorstep delivery during past flood events [97]. Many studies pointed out the citizens will act in their own manner, will take measures when they feel uncomfortable, and will evacuate to a places they feel are safe [104–106]. Round community support by community consultation is very notable for mitigation options. Providing education on flood evacuation to communities can increase the collaboration, support, and acceptance from communities.

4.2.4. Communication and Information

Most importantly, the clear and effective evacuation orders are very sensitive. The content of evacuation, dissemination source, and distribution channel can significantly influence not only the

number of evacuees, but also the urgency during evacuation [107,108]. Nowadays, people gather information through various sources, especially social media. Even though there is a potential negative uses in social media, the benefits using social media for disaster responses and risk reduction could be seen [109,110]. Thus, employment other than those from the government should be taken in consideration and supports it with extra information. Mileti et al. [111] mentioned that people can react in a more appropriate manner when more information available.

## 5. Conclusions

In Thailand, the results of flood risk assessment were rarely shared among citizens [112]. The findings from this study emphasize that low-lying plains are frequent to high flood risk and more than 42,000 citizens struggle with flooding. It is very important and necessary for relevant agencies to prepare and facilitate the effective disaster planning and management; however, unfortunately there are no studies on flood evacuation in the middle CPRB. Thus, we attempted to develop the emergency flood evacuation model for devastating flood events to lessen the risk to people. Based on assumptions in this study, the response to catastrophic flood events in 1995, 2006, and 2011 demonstrated that citizens are capable of evacuating to flood shelters safely.

One role of the model is to foresee the consequences of actions before implementing them. However, evacuation processes are dynamic and rapidly changes due to various uncertainties [113,114]. For example, when road accidents occurs they may provide traffic congestion and previously safe routes may become unsafe. Developing contingency plans in flood evacuation may manage and reduce a fraught and messy evacuation process. Moreover, the behavior of residents in wide-area evacuations is crucial for the success of the evacuation. It is essential for government and emergency agencies to consider these factors to facilitate the best evacuation possible. However, with an available time window from this study, it may be beneficial and provide an opportunity for relevant authorities to implement several measures to improve the effectiveness of evacuation. We believe that results from this study can provide direction or open further discussion on how to draw attention, improve, and create the most effective flood risk management and flood evacuation plan based on the characteristics of the middle CPRB.

**Author Contributions:** S.J. and Y.T. conceived and designed this study; S.J. performed the data collection, modeling work, and primary analysis of the results; Y.T. supervised the methodology, the data analysis, and conclusion; Both authors finalized the research article together.

**Funding:** This research received no external funding.

**Acknowledgments:** The authors are grateful to the Royal Irrigation Department, Electricity Generating Authority of Thailand, Department of Provincial Administration, Department of Highways, The Treasury Department, Department of Disaster Prevention and Mitigation and other agencies for supplying the required data for the study.

**Conflicts of Interest:** The authors declare no conflicts of interest.

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
