# Peer review of "Available Flood Evacuation Time for High-Risk Areas in the Middle Reach of Chao Phraya River Basin"

_water, doi:10.3390/w10121871_

Round 1

Reviewer 1 Report

1. This research is to study the flood evacuation time for high-risk areas according to the flood hazard and social vulnerability maps generated by fuzzy AHP and fuzzy logic. It’s an interesting task and the whole structure of manuscript including theory and analysis is complete and acceptable.

2. The manuscript referred the previous research results of hydrodynamic model as the basis of flood hazard map. The previous study seems using DEM with 30m*30m cells, but 50m*50m grids are used in this study. Why? It would be more precise using a smaller grid, doesn’t it?

3. Another important result is fuzzy AHP results. Authors should clarify that the fuzzy AHP is proposed in this study or previous study. The background of expert group was not mentioned and the operation process was also not illustrated in manuscript.

4. The final important point is the Emergency Flood Evacuation Model. What is the theory of Emergency Flood Evacuation Model? The simulated process? These are not described in this study.

5. The emergency flood evacuation model was used to calculate evacuation time while the flood was occurring or before flooding. The flooding process is dynamic and the flooding are would affect the evacuation behavior. The present study just considered the flood hazard map or considered both flooding depths and the increasing areas with time.

6. All figures are not clear enough to identify the detailed information.

Author Response

First of all, authors would like to thank reviewer to give valuable comments and suggestions to significantly improve this paper.

Reviewer 2 Report

This is for the flood risk map for evacuation time in the middle of Chao Phraya river basin. The study has valuable flooding information and evacuation. However, the following comments could be answered in the manuscript for the clarity.

1. Please explain the reason to classify the low hazard zone, low-medium hazard zone, Medium zone, etc. In Table 3, the percentage of zones were various from 2.87% to 41.02%. The differences seem too big.

2. Please explain the meaning of the calculated weightings of factors. For example, population density is twice important than census population and census housing?

3. Please explain about flood warning system in the study region. What is the average evacuation time after flood warning alert? In Figure 7, evacuation travel times are more than 150 min. Since the study area is low-lying area right downstream of the steep mountains, the evacuation time should shorter than the possible flood travel time after flood warning.

4. Please provide the previous studies for the structural and non-structural flood measures in the region and compare to the proposed measures in the study.

Author Response

(The authors gave the same response as above.)

Round 2

Reviewer 1 Report

The form of tables should be revised to fit the standard of this journal.

Author Response

I would like to thank the editor, associate editor, and reviewers for comments and suggestions. These have helped me to significantly improve the manuscript. My explanations and answers are also shown with yellow highlights.
